# Development of a High-Efficacy Reprogramming Method for Generating Human Induced Pluripotent Stem (iPS) Cells from Pathologic and Senescent Somatic Cells

**DOI:** 10.3390/ijms21186764

**Published:** 2020-09-15

**Authors:** Naomichi Tanaka, Hidemasa Kato, Hiromi Tsuda, Yasunori Sato, Toshihiro Muramatsu, Atsushi Iguchi, Hiroyuki Nakajima, Akihiro Yoshitake, Takaaki Senbonmatsu

**Affiliations:** 1Department of Cardiology, International Medical Center, Saitama Medical University, Saitama 350-1298, Japan; ntanaka@saitama-med.ac.jp (N.T.); koma0726@saitama-med.ac.jp (H.T.); toshi_m@saitama-med.ac.jp (T.M.); 2Department of Anatomy, Ehime University School of Medicine, Ehime 791-0295, Japan; kato.hidemasa.zz@ehime-u.ac.jp; 3Department of Preventive Medicine and Public Health, Keio University School of Medicine, Tokyo 160-8582, Japan; yasunori.sato@keio.jp; 4Department of Cardiovascular Surgery, International Medical Center, Saitama Medical University, Saitama 350-1298, Japan; iguchi27@saitama-med.ac.jp (A.I.); nakajim@saitama-med.ac.jp (H.N.); akihiro@saitama-med.ac.jp (A.Y.); 5Research Administration Center, Saitama Medical University, Saitama 350-0495, Japan

**Keywords:** iPS cell, myocardial fibroblast, myofibroblast, senescence, pathologic state, TGF-beta

## Abstract

Induced pluripotent stem (iPS) cells are a type of artificial pluripotent stem cell induced by the epigenetic silencing of somatic cells by the Yamanaka factors. Advances in iPS cell reprogramming technology will allow aging or damaged cells to be replaced by a patient’s own rejuvenated cells. However, tissue that is senescent or pathologic has a relatively low reprogramming efficiency as compared with juvenile or robust tissue, resulting in incomplete reprogramming; iPS cells generated from such tissue types do not have sufficient differentiation ability and are therefore difficult to apply clinically. Here, we develop a new reprogramming method and examine it using myofibroblasts, which are pathologic somatic cells, from patient skin tissue and from each of the four heart chambers of a recipient heart in heart transplant surgery. By adjusting the type and amount of vectors containing transcriptional factors for iPS cell reprogramming, as well as adjusting the transfection load and culture medium, the efficiency of iPS cell induction from aged patient skin-derived fibroblasts was increased, and we successfully induced iPS cells from myocardial fibroblasts isolated from the pathologic heart of a heart transplant recipient.

## 1. Introduction

One of the most promising technologies in regenerative medicine is induced pluripotent stem (iPS) cells, which are artificial pluripotent stem cells induced via the epigenetic silencing of somatic cells by the Yamanaka factors, i.e., the four transcription factors Oct4, Sox2, Klf4, and c-Myc [1]. Although iPS cells were first invented over a decade ago, and many excellent studies have been conducted on these cells, there are still no full-scale clinical applications of this expensive technology.

Several reprogramming techniques have been developed for generating iPS cells via transferring the genes for the four Yamanaka transcriptional factors to somatic cells. Approaches using lentivirus or retrovirus are highly efficient stable methods for generating iPS cells [2,3]. However, because there is a possibility that the transfected genes may be inserted into the genome of the host cell, generating iPS cells in this manner limits their clinical applicability. To avoid potential gene insertion into the host genome, “non-integration” methods have been developed, such as Sendai virus transfection, episomal vector transfection, and mRNA transfection [4,5,6]. Each method has its advantages and disadvantages and is selected based on the researchers’ preferences and environment. Regarding the future application of iPS cells, the most useful and cheap technique will eventually be used in combination with gene-free and small-molecule methods [7].

Recent studies have focused on generating human iPS cells from senescent somatic cells [8]. The percentage of people aged over 60 years in the global population is estimated to reach 21.8% by 2050 [9]. Common age-related diseases include diabetes, cancer, and cardiovascular disease; consequently, an increase in patients with these conditions is considered to be a particularly important social matter in a super-aging society. Advances in iPS cell reprogramming technology could allow aging or damaged cells to be replaced by the patient’s own rejuvenated cells; therefore, the clinical application of iPS cell reprogramming technology may be a solution to the problem of age-related degenerative diseases.

Although iPS cell technology is theoretically able to replace autologous cells with a patient’s own rejuvenated cells, senescent or pathologic tissue has a relatively low reprogramming efficiency compared with juvenile or robust tissue, resulting in incomplete cell reprogramming. Until this challenge is overcome, the insufficient differentiation ability of iPS cells generated by current methods impedes their clinical application. There are several possible causes for the low reprogramming efficiency of these cells. Lapasset et al. reported that by using a cocktail containing genes for the four Yamanaka factors along with NANOG and LIN28 for iPS cell reprogramming, they achieved iPS cell generation from both proliferating fibroblasts and senescent fibroblasts, with similar generative efficiency [10]. However, since both Nanog and Lin28 promote tumor formation, an iPS cell generation method that does not require these two factors would be preferable. By improving the Sendai virus transduction efficiency, Vosough et al. successfully generated iPS cells from senescent somatic cells [11]. These results suggest that senescent somatic cells may have a specific gene that prevents transfection with the transcription factors necessary for generating iPS cells, i.e., the reprogramming of senescent cells is generally not efficient via the conventional method, so it is necessary to establish an experimental method that has high induction efficiency, even in senescent or pathologic cells, leading to the generation of many iPS cell colonies. This situation is due to the occurrence of genomic point mutations and gene copy mutations during the generation of iPS cells, which cannot be improved scientifically but must be mixed in as a probability theory.

There are two major pathways of cellular metabolism, oxidative phosphorylation (OXPHOS) and glycolysis. Most somatic cells depend on OXPHOS because this metabolism produces 18 times more ATP compared with glycolysis, whereas pluripotent stem cells depend on glycolysis despite its less efficient ATP production. Because mammalian embryos reside in a hypoxic environment prior to implantation, during the early stages of embryonic development, there is a metabolic shift from OXPHOS to glycolysis, and oxidative metabolism is fully reinstituted only after implantation. Although the cellular metabolism of iPS cells is not identical to that of embryonic stem (ES) cells, it is like that of ES cells in that it acts via the glycolysis system [12,13]. Thus, when somatic cells are reprogrammed to generate iPS cells, conversion from OXPHOS to glycolytic metabolism occurs. Cacchiarelli et al. reported that there are distinct waves of gene network activation followed by the activation of representative pluripotent genes during reprogramming [14]. Zhang et al. found that iPS cells generated from aged tissue fail to suppress OXPHOS, which results in high levels of reactive oxygen species, triggering the DNA damage response [15]. These results suggest that metabolism is either directly or indirectly involved with every aspect of cell function. Furthermore, when aged or pathologic somatic cells are used for iPS cell reprogramming, even if iPS cells can be successfully generated, the resulting cells may not have sufficient pluripotency.

Here, we developed a new reprogramming method for generating iPS cells using senescent fibroblasts of aged patients who had heart disease and then examined it using pathologic somatic cells from a heart obtained from a heart transplant recipient. After being sorted from patient skin tissue and from each of the four heart chambers, pathologic fibroblasts were cultured, then subjected to reprogramming to generate human iPS cells. We assessed whether this approach could achieve a highly efficient reprogramming method applicable to even poor-quality somatic cells, such as pathologic or aged cells.

## 2. Results

Fibroblasts were sorted from the dermal tissue of three elderly patients (Table 1). Examination of the cells by microscopy revealed that their appearance was like that of normal fibroblasts; they were all positive for β-galactosidase (Figure 1a), indicating that these cells were senescent [16]. However, the degree of β-galactosidase in D_3_ appeared to be clearly low. The fibroblasts from the three patients all had significantly lower proliferation levels compared with commercially available human fibroblasts (Figure 1b). The genetic characteristics of the three groups of senescent fibroblasts were assessed by using digital PCR [17]. TP53, CDKN1A, and CDKN2A are representative markers of senescence. Despite all three being over 70 and all positive of β-galactosidase, the levels of CDKN1A of D_1_ and D_2_, but not D_3_, were significantly higher than that of the control (Figure 1c). ACTA2 (encodes α-smooth muscle actin (α-SMA)) denotes pathology [18]. The levels of α-SMA of D_1_ and D_3,_ but not D_2_, were significantly higher than that of the control (Figure 1c). These results indicate that the patient-derived cells were not normal.

We subjected subsets of fibroblasts from each of the three patients to two kinds of our novel iPS cell reprograming protocols (protocol 1 and protocol 2). In protocol 1, OCT3/4, a short hairpin p53, SOX2, KLF4, LIN28, L-MYC, Glis-1, and Tet-1 were transfected to the fibroblasts in the culture medium with TGFβ. Meanwhile, in protocol 2, they were transfected in the culture medium containing SB431542 but not TGFβ (Figure 2). We then confirmed that the induced colonies were composed of iPS cells by the microscopy, conducting TRA-1-60 immunofluorescent staining and alkaline phosphatase staining [19]. Although, using protocol 1, iPS cell generation was succeeded in only D_3_ by the microscopy, alkaline phosphatase staining of iPS cell colonies was positive in all fibroblasts. Protocol 2 led to iPS cell generation in all fibroblasts by both the microscopy and alkaline phosphatase staining (Figure 3 and Figure 4a,b). The number of iPS cell colonies of protocol 2 was significantly higher than those of protocol 1 in D_1_ and D_3_. In both protocols, the number of iPS cell colonies of D_3_, which did show a pathological condition but not aging, was significantly higher than the others (Figure 4c).

To further test the efficacy of our new protocol, we applied it to myocardial fibroblasts from all four chambers of the heart from a heart transplant recipient who suffered from dilated cardiomyopathy and had bilateral heart failure (Table 1). The number of samples was set to five because fibroblasts were obtained from each of four chambers of the recipient heart and the skin tissue of the patient. We first examined the morphology and gene characteristics of the myocardial fibroblasts from each of the four chambers and dermal tissue. Although these myocardial fibroblasts were morphologically similar to dermal fibroblasts, they all exhibited extremely high β-galactosidase expression (Figure 5a). The dermal fibroblasts and the cardiac fibroblasts of LA showed that pathological and aging markers were normal. The cardiac fibroblasts of RA and RV highly expressed TP53 and CDKN1A, which indicated these cells were in senescence; however, the pathological marker of them was normal. The cardiac fibroblasts of LV highly expressed α-SMA, TP53, and CDKN1A, and this was consistent with the patient’s severe condition (Figure 5b). iPS cell induction from all fibroblasts using protocol 2 was successful; meanwhile, no iPS cell induction occurred using protocol 1 (Figure 6 and Figure 7).

## 3. Discussion

Here, we describe the establishment of a highly efficient method for inducing human iPS cells. This protocol was successfully implemented both in dermal tissue from three elderly patients with heart disease and in pathologic heart tissue from a heart transplant recipient that was removed during the patient’s heart transplantation surgery. The dermal fibroblasts from the patients with heart disease had lower proliferation levels compared with normal fibroblasts, and they were β-galactosidase positive, which indicates senescence [16]. Cells like these typically transform into myofibroblasts expressing high levels of α-SMA, which is a representative marker of myofibroblasts, and their iPS cell induction efficiency with the standard iPS cell induction method is extremely low [20]. By adjusting the type and amount of vectors containing transcriptional factors for iPS cell reprogramming, and by changing the composition of the culture medium to an optimal environment for transformation from somatic cells to undifferentiated cells, we successfully obtained more efficient iPS cell induction in this cell type. Using our new method, particularly using protocol 2, we induced human iPS cells from pathologic myocardial fibroblasts derived from the heart of a heart transplant recipient.

iPS cell induction using residual surgical specimens fits the original potential of iPS cell technology [21]. In a super-aging society, iPS cell reprogramming technology that can induce rejuvenated autologous cells may provide a solution to the problem of age-related degenerative diseases by replacing the aged and pathologic cells in patients with these conditions. Because cellular senescence has hindered efficient iPS cell reprogramming, developing a method capable of efficiently inducing iPS cells from aged somatic cells has been challenging. Here, we aimed to improve the efficiency of iPS cell reprogramming using aged or pathologic cells.

As shown in Figure 1, the proliferative activity of dermal fibroblasts obtained from the skin tissue of elderly patients was lower than that of control cells. Although we used a fibroblast-specific medium containing several growth factors and tried various modifications of the standard protocol in an effort to increase the proliferative power of these cells, in most cases, iPS cells were not induced using standard protocol, and even when they were induced, the reprogramming in these cells was incomplete.

In consideration of future clinical applications for iPS cells, a non-integration method was incorporated into our protocol to prevent the four transfected Yamanaka factor genes from integrating into the host genome. Of the several available non-integration methods, such as episomal vectors, Sendai virus, and RNA [22], the Sendai virus method is thought to yield efficient and highly reliable reprogramming, and the resulting viral sequences eventually disappear in most cell lines at higher passages [23]. Vosough et al. reported that reprogramming using Sendai virus applied with hydrodynamic pressure via centrifugation led to a dramatic improvement in the transduction efficiency of centenarian cells [11]. This approach was designed based on the hypothesis that the cause of insufficient transduction efficiency may be a significantly reduced amount of reprogramming factor being delivered to the cells [24]. However, Sendai virus reprogramming kits are supplied by only one commercial vendor, so we designed our iPS cell reprogramming method using episomal vectors, which permit a high degree of freedom in the selection and transfected amounts of vectors [22].

To date, various chemical transfection techniques based on lipofectamine and commercially available electroporation methods have been tested on fibroblasts. Because the best transfection results for fibroblast cell lines were generated by applying Lonza Japan’s Nucleofector 2b, which is an electroporation device, in combination with the basic Nucleofector™ kit for primary mammalian fibroblasts VPI-1002 using the Nucleofector 2b protocol U-020, we incorporated these methods into our iPS cell induction protocol (data not shown) [25,26]. There is a trade-off relationship between transfection efficiency and cell survival rate; therefore, the use of similar transfection protocols V-007, U-007, U-009, and U-014 could be considered for future applications depending on the patient’s condition and age.

The Yamanaka factor cocktail consists of four transcriptional factors, Oct4, Sox2, Klf4, and c-Myc1. However, we were unable to successfully reprogram dermal fibroblasts from elderly patients when using only the Yamanaka factors. As an alternative method, Yu et al. reported a new transcription factor cocktail, containing the genes for Oct4, Sox2, Nanog, and Lin28; this cocktail was sufficient to reprogram human somatic cells into pluripotent stem cells that exhibit the essential characteristics of embryonic stem (ES) cells [3]. Furthermore, Lapasset et al. discovered a six-factor reprogramming cocktail, which contains genes for the four Yamanaka factors plus those for Nanog and Lin28; this cocktail showed a high reprogramming efficacy for generating iPS cells from senescent fibroblasts [10]. However, this cocktail is not ideal because Nanog and Lin28 were previously found to promote tumor formation [27,28]. Our novel protocol employed six transfection vectors: episomal vector pCXLE-hOCT3/4-shp53F, which is a fusion of the human transcription factor gene OCT3/4 and a short hairpin p53, to prevent tumor formation; episomal vector pCXLE-HSK, which is a fusion of the human transcription factor genes SOX2 and KLF4; episomal vector pCXLE-HUL, which is a fusion of human LIN28 and human L-MYC instead of C-MYC, for more efficient iPS cell induction; pCXLE-hGLIS-1, which contains the human transcription factor gene GLIS-1; pCXLE-hTET-1, which contains the human demethylase gene TET-1; and pCXWB-EBNA1, which is a non-replicating episomal expression of EBNA1. Three of these vectors, pCXLE-hOCT3/4-shp53F, pCXLE-HSK, and pCXLE-HUL, are used in the protocol published by the Center for iPS Cell Research and Application, Kyoto University. Despite its connection with tumor formation, human LIN28 was employed in this method because Lin28, which is known to improve reprogramming efficiency, was found to prevent the reversion of iPS cells to somatic cells and to increase the efficiency of iPS cell generation, and it is considered necessary for iPS cell induction from aged cells and pathologic cells [29]. Gli-similar transcription factors (Glis) belong to the group of Krüppel-like zinc-finger transcription factors, and three GLIS genes (GLIS1–3) have been identified [30]. Yasuoka et al. reported that Glis1 is significantly expressed in early mouse embryos and that when reprogramming human and mouse fibroblasts to iPS cells, the use of modified Yamanaka factors (Oct4, Sox2, Klf4, and Glis-1 instead of c-Myc) decreases tumorigenicity [31]. Additionally, Yoshioka et al. reported that Yamanaka factor RNAs with Glis-1 RNA that were purified from an RNA-replicative vector yielded high-efficiency iPS cell reprogramming from older adult human cells [32]. Therefore, in our novel method, the GLIS-1 gene was added as a transcription factor in place of the gene for the Yamanaka factor c-Myc.

DNA methylation is one of the most well-characterized epigenetic modifications and is essential for suppressing gene expression and maintaining genomic stability in many organisms [33]. Ito et al. reported that Tet-1, which is a DNA demethylase, contributes to the differentiation of the inner cell mass at the blastocyst stage and regulates the maintenance of ES cells by altering the DNA methylation status [34]. Furthermore, Olariu et al. found that TET1 could replace OCT4 in the iPS cell reprogramming Yamanaka cocktail and that DNA methylation is the key to regulating pluripotency genes [35]. Because Tet-1 may also evoke the induction of Dnmt3b expression upon transition to the epiblast stage, TET-1 was included in the final set of six genes to be transfected in our protocol. Although there were several other candidate genes that could have been additionally transfected, there is a trade-off relationship in gene transfection between the vector amount and the host cell viability. Transfection with a large amount of vector requires a high electroporation load for a sufficient amount of vector to be transferred. In high-load electroporation, the host cell damage is substantial, and our attempts at optimizing the transfection method alone when using more genes and the corresponding high amounts of vector did not give a sufficient iPS cell induction effect on aging pathologic cells. As result of these improvements, such as adjusting the type and amount of vectors containing transcriptional factors, as well as adjusting the transfection load, protocol 1 was developed. It succeeded in achieving iPS cell generation in fibroblasts of D_3_ that showed a pathological condition but not aging.

After being sorted, the cells cultured from the left ventricle (LV) of a heart transplant recipient were mainly myofibroblasts that expressed high levels of α-SMA. This characteristic reflects the pathologic condition of the heart mesenchymal tissue from the patient. Despite being 58 years old, the cardiac fibroblasts of LV also showed senescence. This indicated that the cardiac fibroblasts of LV converted to the myofibroblasts being aging. Myofibroblasts occur at a converging spot of mesenchymal cells, resulting from acute or chronic inflammation caused by stimulation with TGF-β, Ang II, and cytokines. These cells are key players in organ fibrosis. Interestingly, the myofibroblasts from our patient’s heart tissue did not differentiate into human iPS cells at all when treated using protocol 1, and most of the cells lost their proliferative capacity and became senescent. Cacchiarelli et al. analyzed the genetic alterations in monoclonal iPS cells resulting from transfection with the Yamanaka factors. The results indicate that, in mesenchymal cells, the expression levels of representative genes were reduced for about a week immediately after iPS cell induction [14]. However, when iPS cells were induced from myofibroblasts, the initial changes did not occur, and the cells eventually aged and died. Yamasaki et al. found that TGF-β is important as an essential factor in the culture medium for maintaining pluripotency [36]. TGF-β promotes myofibroblast proliferation; consequently, the initial decrease in the expression of mesenchymal genes after gene transfection was suppressed in the presence of TGF-β [18]. Because of this, there was a high possibility that iPS cell induction would not be achieved if TGF-β was present during the early stages of reprogramming. Additionally, SB431542 has been reported to enhance the efficiency of iPS cell induction by promoting Oct4 expression [37,38]. Therefore, in our novel iPS cell induction protocol 2, the initial culture medium used during the first 5 days post-transfection excluded TGF-β and included the selective TGF-β inhibitor SB431542. We successfully increased the efficiency of iPS cell induction from myofibroblasts by constructing a culture environment in which the downregulation of major mesenchymal genes in the early stage of the transfection protocol is likely to occur.

In addition, the main metabolism of somatic cells is OXPHOS, and the main metabolism of undifferentiated cells is glycolysis [39]. Although the standard protocol for iPS cell induction uses a 5% O_2_ environment, because somatic cell properties remain strong during the early stage of iPS cell induction, we implemented 20% O_2_ conditions over the time period during which TGF-β was excluded from the culture medium.

As a result of adjusting the types and amounts of vectors containing transcription factors for iPS cell reprogramming, the transfection load, and the culture medium, the efficiency of iPS cell induction from aged patient skin-derived fibroblasts was increased. In the fibroblasts of D_3_ that were pathological but did not show senescence, iPS cell generation was succeeded using both protocol 1 and protocol 2, and the number of colonies in protocol 2 was significantly higher than that of protocol 1. On the other hand, in the fibroblasts of D_1_ that showed a pathological condition and senescence, although protocol 2 led to iPS cell generation, little could be done using protocol 1. It is also possible that the samples obtained are not only from an elderly age but from tissues with pathological conditions. Taken together, since these results indicate that pathological conditions and aging definitely influence iPS cell induction, it was necessary to find out how to improve the iPS induction efficiency in cells with aging and a pathological condition. Furthermore, we also successfully induced iPS cells from myocardial fibroblasts sorted from the pathologic and aged heart of a heart transplant recipient.

Since all the improvements have their respective effects, it is difficult to list the most important improvements. It is considered that the removal of TGF-β and addition of SB431542 in the culture medium at the early stage of induction is highly effective, because protocol 2 showed a high efficacy compared with protocol 1. Aging or pathological cells have diminished proliferative capacity, as mentioned in the result of Figure 1. Attenuation of mesenchymal genes occurs for about 5 days after gene transfer. However, since TGF-β proliferates pathological fibroblasts, such as myofibroblasts, the attenuation of early mesenchymal genes is suppressed in culture medium with TGF-β. Thus, conversion into undifferentiated cells is suppressed, resulting in the aging or death of cells.

Finally, we developed a highly efficient method for inducing human iPS cells from senescent and pathologic somatic cells using episomal vector transfection, which is the most versatile non-integration method. The optimization of all combinations is important for obtaining an effective protocol. Some patients may need some modification; however, we believe that a modification based on our method will provide sufficient iPS cell induction.

## 4. Limitation

Since heart transplantation was not a common operation, only one case was presented for the study. Although the number of the patient was one, the number of samples was set to five because fibroblasts were obtained from each of the four chambers of the heart and the skin tissue of the patient. Therefore, these samples were used as cardiac fibroblasts with the most advanced pathologic state. The patient was 58 years old and his underlying disease was dilated cardio myopathy. He had a severe left heart failure resulting in a high expressive level of α-SMA in the left ventricle. It indicated that the cardiac fibroblasts of the left ventricle converted to myofibroblasts. α-SMA of the other four samples showed a normal level. Surprisingly, despite the patient being 58 years old, many aging genes were highly expressive. Although we succeeded in iPS cell generation from all samples using protocol 2, for the future, we would like to increase the number of patients and demonstrate that the new protocol we developed is highly efficient for generating iPS cells using pathological somatic cells.

## 5. Materials and Methods

### 5.1. Cell Culture of Dermal Fibroblasts and Myocardial Fibroblasts

KBM (Kohjin bio medium) Fibro Assist was purchased from Kohjin Bio Co., Ltd. (Saitama, Japan). Penicillin-Streptomycin was purchased from Nacalai Tesque Co., Ltd. (Kyoto, Japan). L-ascorbic acid was purchased from FUJIFILM Wako Pure Chemical Corporation (Osaka, Japan). Specimens were skin tissue that was peeled off during trimming at drain tube extubation during Maze surgery and heart tissue from each of the four chambers of a heart transplant recipient during heart transplantation surgery. A total of eight patient specimens were collected (Table 1). Human foreskin fibroblast cells (HFFCs) were purchased from Cellular Engineering Technologies (Coralville, IA, USA) (Product code: HFFC-500). Using the explant method, KBM Fibro Assist + 20% FBS + Penicillin-Streptomycin (100 IU/mL penicillin and 100 µg/mL streptomycin) + L-Ascorbic Acid (1 µg/mL) culture medium was used to grow fibroblasts.

### 5.2. Cell Counts

Fibroblasts (2.0 × 10^5^/well) were seeded on a six-well tissue culture plate and observed with a Keyence (Osaka, Japan) BZ-X710 microscope. Immediately after the cells were seeded, still images at six points per well were taken at 1-h intervals. The number of cells in the images at 48 h post-seeding was measured using the Keyence BZ-X710 macrocell count mode.

### 5.3. Transfection

pCXLE-hOCT3/4-shp53-F, pCXLE-hUL, pCXLE-hSK, pCXLE-hGLIS1, and pCXWB-EBNA1 were purchased from Addgene. pCXLE-hTET-1 was constructed. iMATRIX-511 was purchased from Nippi^®^ (Tokyo, Japan). Fibroblasts (1.0 × 10^6^) from each tissue sample were prepared, then gene transfection for iPS cell reprogramming was performed with the electroporation device Nucleofector 2b (Lonza Japan^®^, Tokyo, Japan) using the U-020 protocol, VPI-1002 transfection kit (Lonza Japan^®^), 1.2 µg each of pCXLE-hOCT3/4-shp53-F, pCXLE-hUL, pCXLE-hSK, pCXLE-hGLIS1, and pCXLE-hTET-1, and 1.0 µg of pCXWB-EBNA1. To reduce the possibility of the induction efficiency being decreased by the entry of a nick when using the stocked vectors, we irradiated the vectors before use. After electroporation, the cells were seeded on 100-mm dishes that had previously been coated with collagen and 0.125 µg/cm^2^ of iMATRIX-511. In protocol 1, after day 1 post-transfection, they were cultured in an incubator set at 37 °C with 5% O_2_. In protocol 2, from day 1 to day 5 post-transfection, the cells were cultured in an incubator set at 37 °C with 20% O_2_; after day 5 post-transfection, they were cultured in an incubator set at 37 °C with 5% O_2_.

### 5.4. Reprogramming Medium Protocol

The two reprogramming protocols used for iPS cell differentiation are depicted in Figure 2. FGF2 was purchased from PeproTech (Rocky Hill, NJ, USA). SB431542 was purchased from FUJIFILM Wako Pure Chemical Corporation. In protocol 1, cell culture was performed using TGF-β-containing Essential 8 (Thermo Fisher Scientific^®^, Waltham, MA, USA) immediately after gene transfection. Penicillin-streptomycin (100 IU/mL penicillin and 100 µg/mL streptomycin) was added from day 2 post-transfection. From then onward, the medium was changed every other day. In protocol 2, from day 1 to day 5 post-transfection, a culture medium in which FGF2 (10 ng/mL) and SB431542 (5 µM) was added to Essential 6 (Thermo Fisher Scientific^®^) without TGF-β was used. Penicillin-streptomycin (100 IU/mL penicillin and 100 µg/mL streptomycin) was added from day 2. After day 6, the culture medium was switched to Essential 8.

### 5.5. Immunofluorescent Staining

A Cellular Senescence Detection Kit-SPiDER-βGal containing Hoechst33342 (nuclear stain) was purchased from Dojindo^®^ (Kumamoto, Japan). An Anti-TRA-1-60 antibody, anti-SSEA4 antibody, and alkaline phosphatase stain kit (AP Staining Kit, Red-Color) was purchased from Funakoshi^®^ (Tokyo, Japan). Fluorescent staining of each fibroblast sample was performed by using the Cellular Senescence Detection Kit-SPiDER-βGal. This kit contains Hoechst33342; double staining was performed in accordance with the kit protocol. Fluorescent staining with anti-TRA1-60 and anti-SSEA4 antibodies was performed on induced iPS cell colonies from day 21 to day 35 post-transfection. At day 36 post-transfection, staining with alkaline phosphatase was performed, and the number of iPS cell colonies was evaluated.

### 5.6. Digital PCR

Digital PCR uses a new method to detect and quantify nucleic acids, and estimates the absolute number of molecules through statistical methods. This technique provides absolute quantitation detection independent of the number of amplification cycles. Therefore, it is an approach different from conventional real-time PCR and we performed PCR using this. An RNA extraction kit (NucleoSpin^®^ RNA) was purchased from Takara Bio (Shiga, Japan). A cDNA synthesis kit (PrimeScript™ RT reagent Kit) was purchased from Takara Bio. ddPCR GEX HEX Assays that contain specific primers for digital PCR were ordered from Bio-rad (Hercules, CA, USA) (ddPCR™ GEX Assay: ANPEP, *Homo sapiens*; ddPCR™ GEX Assay: ACTA2, *Homo sapiens*; ddPCR™ GEX Assay: SNAI2, *Homo sapiens*; ddPCR™ GEX Assay: VIM, *Homo sapiens*; ddPCR™ GEX Assay: TP53, *Homo sapiens*; ddPCR™ GEX Assay: CDKN1A, *Homo sapiens*; and ddPCR™ GEX Assay: CDKN2A, *Homo sapiens*). RNA was extracted from fibroblasts, then cDNA was synthesized from the RNA. The cDNA was diluted based on the protocol provided by Bio-Rad, then digital PCR was performed. All measured values were corrected using the expression level of GAPDH (ddPCR™ GEX FAM Assay: GAPDH, *Homo sapiens*).

### 5.7. Statistical Analysis

Statistical tests to determine the significance of differences between two groups were performed by using the Student *t*-test; *p*-values of less than 0.05 were taken to indicate significance. Two-way ANOVA was performed where indicated, which was used to examine the interaction and main effects of the two experimental factors. The false discovery rate (FDR) was computed as an adjusted *p* value to account for multiple testing and a cut-off of 5% FDR was used to define differential expression.

### 5.8. Ethics Statement

The protocol was approved by the Ethics Committee of the IRB Committee, Saitama Medical University International Medical Center. Approval numbers are (18–149) and (19–115). The approval dates are (08-09-18) and (04-09-19).

## Figures and Tables

**Figure 1 ijms-21-06764-f001:**
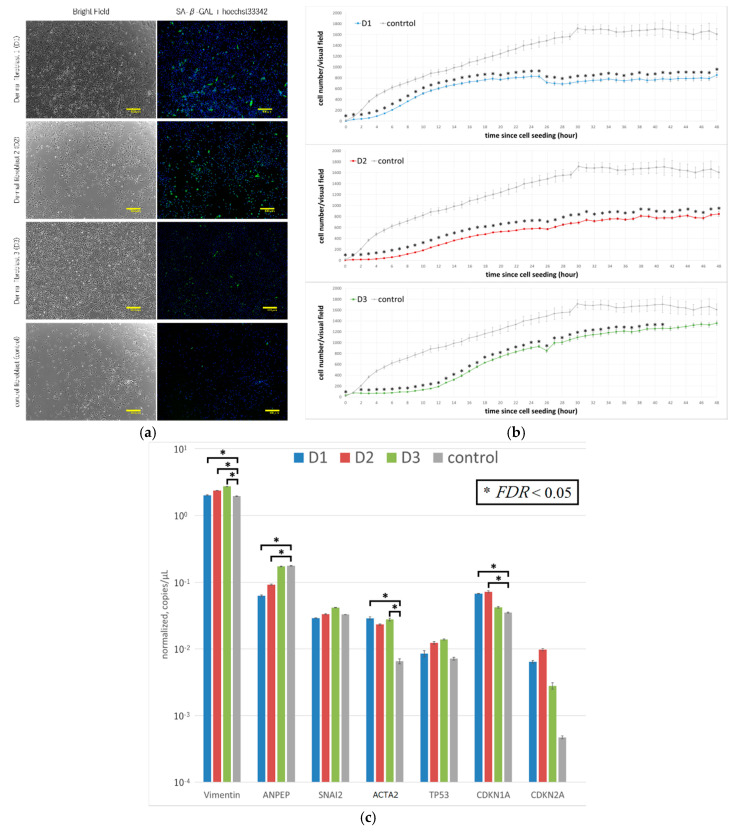
Fibroblasts from the dermal tissue of three elderly patients. (**a**) Left: bright field images of primary cultured dermal fibroblasts derived from three patients. From top to bottom, the cells shown are D1, D2, D3, and control fibroblasts. Right: images of the same cells, immunofluorescently stained for SA-β-Gal and stained with hoechst33342. Scale bars: 400 µm. (**b**) Cell proliferation of fibroblasts from the same three elderly patients. From top to bottom, the graphs show: D1 vs. control, D2 vs. control, and D3 vs. control. The cell number of six points per well was measured at 1-h intervals. The number of cells in the images obtained at 48 h after cells were seeded into a dish was calculated using the Keyence BZ-X710 microscope macro cell count mode. Mean values were plotted on a graph. Results are shown as the mean + SEM (* *p* < 0.05, *n* = 6). (**c**) Expression levels of vimentin, ANPEP, SNAI2, ACTA2, TP53, CDKN1A, and CDKN2A in fibroblasts from the same three elderly patients and in control fibroblasts. Values shown are the mean ± SEM (* *FDR* < 0.05, by two-way ANOVA).

**Figure 2 ijms-21-06764-f002:**
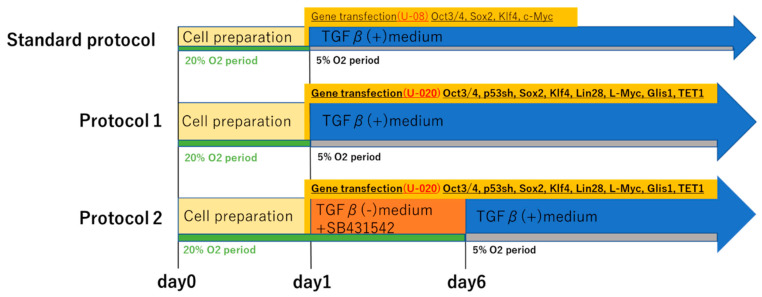
Protocols used for iPS cell reprogramming. Reprogramming protocols for iPS cell differentiation. Top indicates the standard protocol. This follows the protocol published by Center for iPS cell Research and application, Kyoto University. Protocol 1 (middle) and protocol 2 (bottom) are the novel protocol developed in this study. SB431542 is a selective inhibitor of the transforming growth factor beta (TGF-β) type I receptor activin receptor-like kinase ALK5. “TGF-β(+) medium” indicates that TGF-β was included in the reprogramming medium, and “TGF-β(−) medium + SB431542” indicates that the reprogramming medium contained SB431542 but did not include TGF-β. Day 1 was defined as the date of gene transfer. “20% O_2_ period” indicates that cells are incubated in O_2_ 20%, and “5% O_2_ period” indicates that cells are incubated in O_2_ 5%. Each “U-08” or “U-020” indicates the transfection program for the Lonza nucleofector transfection system. “Oct3/4, Sox2, Klf4, c-Myc” or “Oct3/4, p53sh, Sox2, Klf4, Lin28, L-Myc, Glis1, and TET1” are transfected genes in episomal vectors. p53sh indicates short hairpin p53.

**Figure 3 ijms-21-06764-f003:**
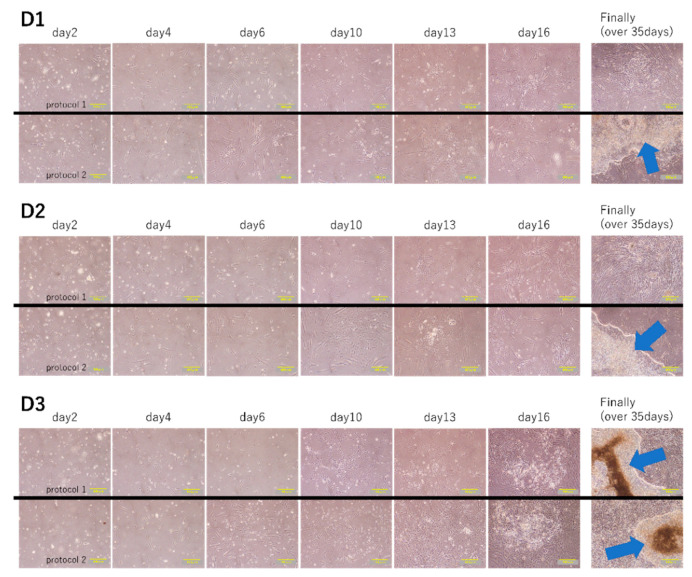
Induction of iPS cell colonies. The process of iPS cell induction in cells from three elderly patients. Gene transfection was performed on day 1. The time post-gene transfer is shown above each pair of images. Cells from D1 (top), D2 (middle), or D3 (bottom) were subjected to protocol 1 (upper row) or protocol 2 (lower row). Arrows indicate the iPS cell colonies that were eventually induced by day 35 post-transfection. Scale bars: 400 µm.

**Figure 4 ijms-21-06764-f004:**
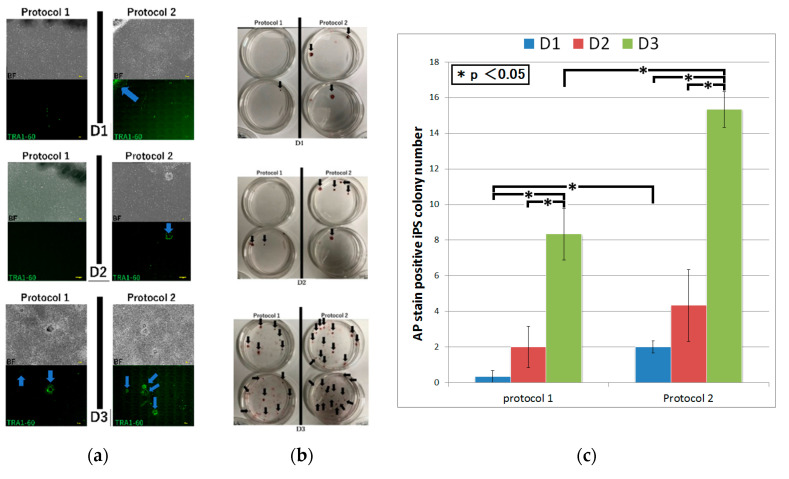
TRA-1-60 and alkaline phosphatase staining of iPS cell colonies induced from the dermal tissue of three elderly patients. (**a**) TRA-1 immunofluorescent staining of iPS cell colonies. Immunofluorescent staining with anti-TRA1-60 antibody was performed on D1 (top), D2 (middle), and D3 (bottom) cells subjected to protocol 1 (left) or protocol 2 (right) for iPS cell induction at day 35 post-transfection. An integration of 81 continuous fields of the view captured with the Keyence BZ-X710 microscope is shown (observed area: 23.915 mm × 17.936 mm). BF, bright field. Arrows indicate TRA1-60-positive iPS cell colonies. Scale bars: 3.0 mm. (**b**) Alkaline phosphatase staining of iPS cell colonies. Staining with alkaline phosphatase was performed on D1 (top), D2 (middle), and D3 (bottom) cells subjected to protocol 1 (left) or protocol 2 (right) for iPS cell induction at day 36 post-transfection. Arrows indicate alkaline phosphatase-positive iPS cell colonies. (**c**) The number of iPS cell colonies stained by alkaline phosphatase was quantitively analyzed in all fibroblasts. The left side is protocol 1 and the right side is protocol 2. Values shown are the mean ± SEM (* *p* < 0.05, *n* = 3).

**Figure 5 ijms-21-06764-f005:**
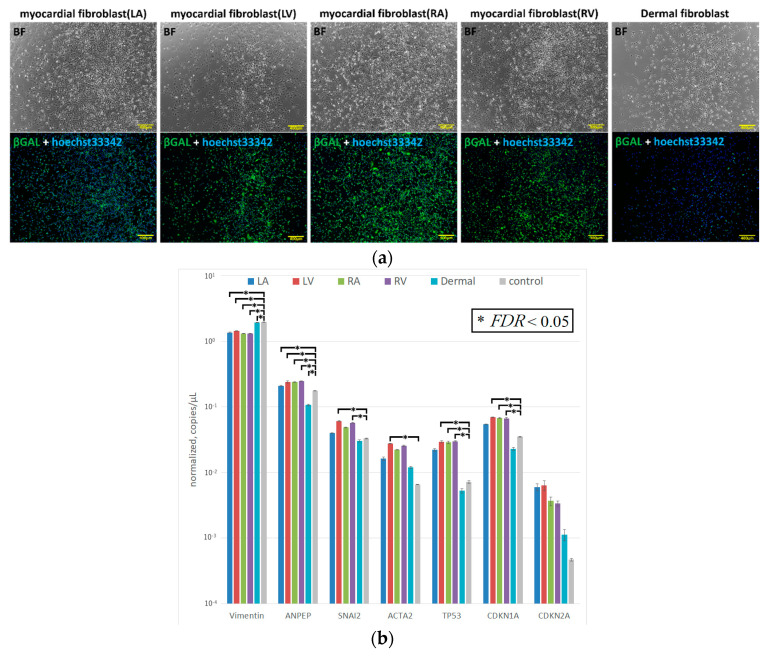
Heart transplant recipient tissue characteristics. (**a**) Bright field (BF; top) and double-stained (subjected to immunofluorescent staining of SA-β-Gal (βGAL) and hoechst33342 staining) images of primary cultured dermal fibroblasts and myocardial fibroblasts derived from a heart transplant recipient. From left to right, myocardial fibroblasts from the left atrium (LA), left ventricle (LV), right atrium (RA), and right ventricle (RV) and dermal fibroblasts are shown. Scale bars: 400 µm. (**b**) Expression levels of vimentin, ANPEP, SNAI2, ACTA2, TP53, CDKN1A, and CDKN2A in myocardial and dermal fibroblasts from a heart transplant recipient. Values shown are the mean ± SEM (* *FDR* < 0.05, by two-way ANOVA).

**Figure 6 ijms-21-06764-f006:**
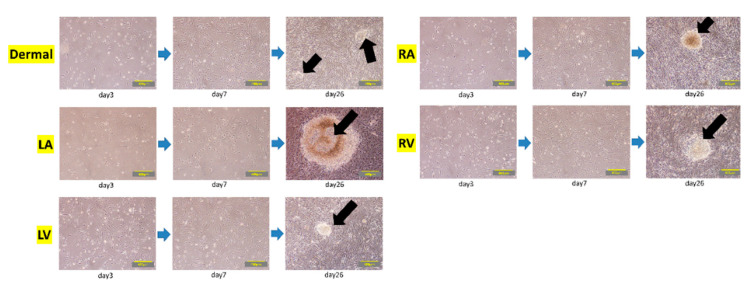
Induction of iPS cell colonies in myocardial fibroblasts from a heart transplant recipient. The process of iPS cell induction in dermal fibroblasts (dermal) and in myocardial fibroblasts from the left atrium (LA), left ventricle (LV), right atrium (RA), and right ventricle (RV) from a heart transplant recipient. Gene transfection was performed on day 1. Images show the cells on day 3, day 7, and day 26 post-transfection. Arrows indicate the iPS cell colonies that were eventually induced. Scale bars: 400 µm.

**Figure 7 ijms-21-06764-f007:**
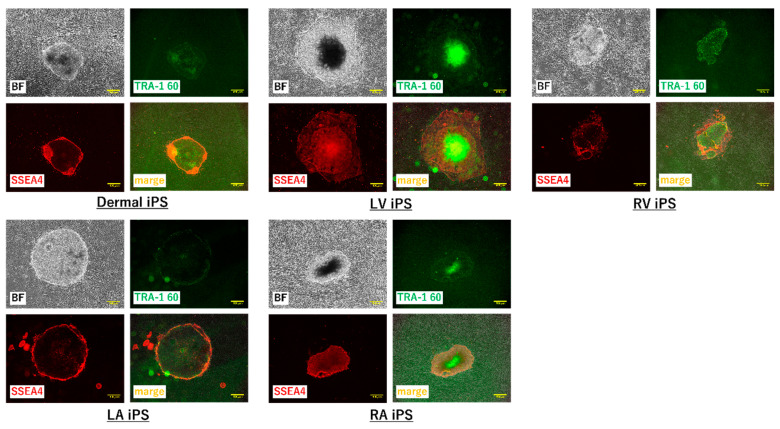
TRA-1-60 and SSEA4 immunofluorescent staining of iPS cell colonies induced from heart transplant recipient myocardial fibroblasts. Bright field (BF; top left), TRA1-60 immunofluorescent staining (top right), SSEA4 immunofluorescent staining (bottom left), and merged (bottom right) images of the iPS cell colonies induced from dermal fibroblasts (dermal) and myocardial fibroblasts from the left atrium (LA), left ventricle (LV), right atrium (RA), and right ventricle (RV) of a heart transplant recipient. Immunofluorescent staining was performed with anti-TRA1-60 antibody and anti-SSEA4 antibody between day 26 and day 35 post-transfection. Scale bars: 400 µm.

**Table 1 ijms-21-06764-t001:** Patient information.

Sample Name	Surgery	Tissue	Age (years)	Sex	Disease
D1	maze	skin	74	male	atrial fibrillation
D2	maze	skin	73	male	atrial fibrillation
D3	maze	skin	74	female	atrial fibrillation
Dermal	heart transplantation	skin	58	male	dilated cardiomyopathy (bilateral heart failure)
LA		left atrium			
LV		left ventricle			
RA		right atrium			
RV		right ventricle			

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
