# Peer review of "Development of a High-Efficacy Reprogramming Method for Generating Human Induced Pluripotent Stem (iPS) Cells from Pathologic and Senescent Somatic Cells"

_ijms, 2020, doi:10.3390/ijms21186764_

Round 1

Reviewer 1 Report

Manuscript is well drafted and the research work is well designed as well as innovative. 

However, there are some corrections to be considered.

  1. Quality of images can be improved, especially the ones in phase contrast.
  2. For Figures 1, 3, 4 & 6: please include scale bar to the images.
  3. what is Digital PCR ? is it like normal RT PCR ? If yes, please mention it in the manuscript.
  4. For references cited some are old, please replace them with latest ones.  (2016-2020)

Reviewer 2 Report

In this study, Tanaka et al. developed an efficient reprograming protocol for generating human iPS cells from senescent fibroblasts and pathological myocardial fibroblasts.

The purpose of the study is very important and the authors optimized several factors: reprogramming factors, electrophoration protocol, TGF-B inhibition, 20% CO2 concentration at the initial reprograming period, etc.

However, it is a big issue that the quantitative analysis of the superiority of the new protocol was not performed in this study. In addition, with regard to the iPS cell generation from pathological myocardial fibroblasts, it had only been performed from one patient and the new protocol was not been confirmed to be generalized. This reviewer has several concerns as below.

Concerns:

  1. As I mentioned above, the quantitative analysis to show the superiority of the new protocol should be performed (Figure 2, 3, 5, and 6).
  2. Line 117: There is no description of the new protocol in advance of the results. Therefore, it is not clear what was modified in the protocol until the end of the discussion. It should be considered in the way it is written to make it easier for the readers to understand.
  3. In this study, the authors modified several factors in the reprogramming methods, but have not investigated which is important. I think that a more detailed study would enable better optimization.
  4. Line 139: As I mentioned above, it is a limitation that generation of iPS cells from pathological myocardial fibroblasts had only been performed in one patient. Further confirmation is needed to prove the efficacy of the new protocol. It should be mentioned in the limitation section.
  5. I understand the clinical importance of making iPSCs from senescent fibroblasts, but what is the clinical application or importance of generating iPSCs from pathological myocardial fibroblasts?
  6. Regarding the statistical analysis in Figure 1c and Figure 4b, multivariate analysis should be performed.
